# The relationship between sleep disorders and postoperative delirium in adult patients: Protocol for an updated systematic review and meta-analysis

Qianli Huang[1,2⊛], Changhui Shao[3⊛], Ling Peng[1], Wei Wei[1*]

**1** Department of Anesthesiology, West China Hospital, Sichuan University, Chengdu, Sichuan, China, **2** Department of Anesthesiology, Chengdu Women's and Children's Central Hospital, School of Medicine, University of Electronic Science and Technology of China, Chengdu, Sichuan, China, **3** Department of Anesthesiology, Chengdu integrated TCM & Western Medicine Hospital, Chengdu, Sichuan, China

⊛ These authors contributed equally to this work.
* weiw@scu.edu.cn

## Abstract

### Introduction

Postoperative delirium (POD) is a common complication after surgery. The association between sleep disorders and the risk of POD has been widely studied. Sleep disorders have emerged as a potential risk factor for POD, but recent studies provide conflicting evidence regarding the existence and the extent of the association. The aim of this systematic review and meta-analysis will be to estimate the association between sleep disorders and the risk of POD in adult patients.

### Methods and analysis

A systematic review will be conducted to estimate the association between sleep disorders and the risk of POD in adult patients. This systematic review protocol follows the Preferred Reporting Items for Systematic Review and Meta-Analysis Protocols (PRISMA-P) statement. Literature searches will be carried out in the PubMed, Embase, CINAHL, and PsycINFO databases from inception until August 2025 without language restrictions. Only observational studies that investigated the association between sleep disorders and POD will be included. Two independent reviewers will be responsible for the selection of studies, extraction of information and risk of bias assessment of the articles. A random effects model will be used to calculate the pooled risk estimates for the outcome. Subgroup analysis will be conducted to explore potential sources of heterogeneity. Publication bias will be estimated by funnel plots and Egger's test. Sensitivity analysis will also be performed to explore the stability of the overall effect size. Also, evidence quality will be assessed. All data will be analysed using Review Manager (V.5.3) and Stata (V.14.0) statistical software.

which permits unrestricted use, distribution, and reproduction in any medium, provided the original author and source are credited.

**Data availability statement:** All relevant data are within the paper and its Supporting Information files.

**Funding:** The research work was funded by a grant from the National Natural Science Foundation of China (grant no. 81971772). The funders had no role in study design, data collection and analysis, decision to publish, or preparation of the manuscript.

**Competing interests:** The authors have declared that no competing interests exist.

## Discussion

The proposed systematic review and meta-analysis will highlight the association of sleep disorders and the risk of POD in adult patients. The findings of this review and meta-analysis will help clinicians consider the sleep disorders to make better health decisions.

**Trial registration**: Prospero registration number: CRD42024604118

## Introduction

Postoperative delirium (POD) is a common and serious postoperative complication, affecting up to 50% of patients in some cohorts [1]. It is associated with a range of adverse outcomes, such as prolonged hospital stays, elevated healthcare costs, cognitive impairment, and higher risk of mortality and morbidity [2–4]. Given the significant adverse outcomes associated with POD, it is crucial for clinicians to identify risk factors and quantify their impact on POD.

Sleep is a complex physiological process that is essential for overall health and well-being. Sleep disorders are highly prevalent during the perioperative period, including conditions such as insomnia, sleep-disordered breathing, and circadian rhythm disruption. Recent studies have shown that 8.8%−79.1% of patients complained about poor sleep quality in the preoperative period [5–7]. Additionally, postoperative sleep disorders are highly prevalent in the first 2 weeks after surgery [8]. In the context of surgical patients, perioperative sleep disturbances may be associated with adverse prognostic outcomes, such as higher incidence of delirium and cognitive impairment, increased postoperative pain intensity, delayed postoperative recovery, increased hospital length of stay, and poor quality of life [9–11].

Extensive research has investigated the relationship between sleep disorders and the risk of POD. An earlier meta-analysis concluded that pre-operative sleep disturbances are likely associated with POD [12]. However, whether postoperative sleep disturbances serve as a risk factor for POD is still unknown. Initially, a meta-analysis conducted by Wang et al. had investigated the association between perioperative sleep disturbance and the risk of POD [13]. Results showed that perioperative sleep disturbances were associated with an increased risk of POD in observational trials, yet no significant association was detected in randomized controlled trials. Meanwhile, this study was limited by high heterogeneity and a small sample size, which may have affected the generalizability and statistical power of the findings. A recent meta-analysis incorporated 16 observational studies which concluded that sleep disturbance may increase the risk of POD [14]. This analysis offered valuable insights but was limited by relatively small sample sizes and the inclusion of only English-language literature.

However, since the publication of the previous meta-analysis, there have been additional studies published with large-scale investigations [15,16]. Considering the

importance of the topic, the aim of this study is to update the current evidence on the relationship between sleep disorders and the risk of POD.

## Methods

### Protocol and registration

This systematic review and meta-analysis was performed in accordance with the Preferred Reporting Items for Systematic Reviews and Meta-Analyses (PRISMA) guidelines [17] and the study protocol was registered in the International Prospective Register of Systematic Reviews (PROSPERO) (registration number: CRD42024604118). The PRISMA-P-checklist is shown in S1 File.

### Ethics and dissemination

As this systematic review is a summary and analysis of existing literature, there is no requirement for ethics approval. The results of the study will be published in a peer-reviewed journal.

### Patient and public involvement

Patients and/or the public were not involved in the design, or conduct, or reporting, or dissemination plans of this research.

### Objectives and outcomes

The sole objective of this study is to determine whether perioperative sleep disorders are associated with an increased risk of POD in adult surgical patients. The study aims to answer the research questions based on PECOS (Population, Exposure, Comparator, Outcomes and Study design) criteria:

What is the association between perioperative sleep disorders and the risk of POD in adult surgical patients?

### PECOS eligibility criteria

- Population: adult patients aged 18 years and older undergoing surgery without pre-existing delirium/neurocognitive disorders before admission.
- Exposure: studies reporting on sleep disorders as exposure variables, defined as those occurring in either the pre- or post-operative period prior to delirium.
- Comparator: the population without exposure to sleep disorders served as a comparator
- Outcome: the occurrence of POD measured by a validated scale, such as the 5th edition of Diagnostic and Statistical Manual of Mental Disorders (DSM-5), the Confusion Assessment Method (CAM), or Confusion Assessment Method for the Intensive Care Unit (CAM-ICU).
- Study design: We will only include observational studies (retrospective or prospective cohort studies, or case-control studies).

**The inclusion criteria are as follows:**

1. Studies investigating the association between sleep disorders and the risk of POD.

2. Studies including participants with and without diagnosed sleep disorders. The included studies should report the subjective or objective diagnostic criteria for sleep disorders. Subjectively measured sleep disorders will be considered

self-reported or obtained by any questionnaires (e.g., Pittsburgh Sleep Quality Index). Data from polysomnography (PSG) or actigraphy will be considered for objective measures of sleep disorders.

**The exclusion criteria are as follows:**

1. Studies without measurement of sleep disorders or POD.

2. Randomized controlled trials (RCTs), cross-sectional studies, review, meta-analyses, letter, editorial, protocol, case reports, or animal experiments.

## Information sources

We will conduct searches in the following databases, including PubMed, Embase, CINAHL, and PsycINFO, for studies published before August 2025 that reported on the association between sleep disorders and postoperative delirium in adult patients. No restrictions will be imposed on language, thereby ensuring the inclusion of all potentially relevant studies. Additional studies will be identified by scanning the reference lists of the eligible studies.

## Search strategy

We will use combinations of the following key terms in our searches: ("sleep" OR "sleep initiation and maintenance disorders" OR "sleep wake disorders" OR "sleep quality") AND ("delirium" OR "emergence agitation" OR "confusion") AND ("postoperative" OR "post surgical" OR "surgery" OR "anesthesia"). Only observational studies, including cohort or case-control designs, will be included. The draft of the search strategy for each database is listed in S2 File.

## Study selection

The records retrieved from the database search will be deduplicated in The Endnote X9 reference management software. Then, two reviewers (QLH and CHS) will independently review the titles and abstracts according to the eligibility criteria. Full-text articles of the remaining records will be assessed against the eligibility criteria. Any disagreements will be resolved by consensus. The reasons for excluding full-text studies will be recorded and reported.

## Data extraction

Two independent reviewers will extract the data from included studies using a Microsoft Excel spreadsheet. Any disagreements in data extraction will be resolved by discussion or consultation with a third reviewer. The following information will be collected: study characteristics (first author, publication year, age, country, number of patients, study design, type of surgery), sleep quality assessment details (sleep quality assessment tool, the timing of sleep disorder assessment, type of sleep disorder assessed, number of patients with good and poor sleep quality), delirium assessment details (delirium assessment tool, number of patients with and without delirium). In cases of missing or unclear data, the corresponding authors will be contacted to obtain the relevant information. If the number of the case group and the control group are not available by literature or mail, the study will be excluded.

## Quality assessment

The Newcastle Ottawa Scale (NOS) assessment will be used to evaluate the quality of the included studies [18]. This assessment tool covers three domains: selection, comparability, and outcome or exposure, with score ranges from 0 to 9. High quality studies will be defined as seven or more scores, moderate quality as five to six scores and low quality as less than five. Any disagreements will be resolved by consensus.

## Publication bias

Publication bias will be visually evaluated by the funnel plot. Any asymmetry detected in funnel plots will imply the presence of bias. Additionally, Egger's tests will be used to assess the severity of publication bias, with $P < 0.05$ considered statistically significant.

## Certainty of evidence

Two independent reviewers (QLH and CHS) will assess the meta-analysis results using the Grading of Recommendations, Assessment, Development and Evaluation (GRADE) approach [19]. We will summarize the quality of evidence across five dimensions, including risk of bias, indirectness of evidence, inconsistency in results, imprecision, and potential publication biases. According to GRADE, the quality of evidence can be qualified in one of four levels: high, moderate, low or very low. Any disagreements will be resolved through discussion or by consulting a third reviewer.

## Data synthesis

In cases where two or more studies are available, reporting on the same outcome measures, a qualitative synthesis will be used to summarize the results of all included studies, otherwise a narrative synthesis will be carried out.

The meta-analysis will be conducted by RevMan 5.3 (Cochrane collaboration, Oxford, UK) and Stata 14(Stata Corp, College Station, TX, USA), considering a two-sided and p value less than 0.05 as statistically significant. Odds ratios (ORs) will be used as common measure of association between sleep disorders and POD; the crude OR and 95% confidence intervals will be estimated for each study, using number of events on each arm where possible. The degree of heterogeneity will be evaluated based on the Cochran's Q statistics and $I^2$ statistics; a value greater than 50% and $p < 0.05$ indicated substantial heterogeneity between studies. Owing to differences in study design and settings, we used the random-effects model [20] to pool all odds ratios. If a study qualifies for inclusion in the systematic review but doesn't supply enough data for the meta-analysis, we'll also perform a systematic review along with a descriptive analysis.

When substantial heterogeneity is detected, we will investigate sources of heterogeneity by using meta-regression analysis or subgroup analysis. Subgroup analysis will be stratified by variables, such as study design (retrospective, prospective), mean population age (greater than 65, less than 65), sleep disorder type (i.e., insomnia, poor sleep quality, obstructive sleep apnoea, and unspecified sleep disorder), timing of sleep disorder (preoperative, postoperative before delirium), study country (China, outside China), and type of surgery (cardiac surgery, non-cardiac surgery) to determine whether other factors affected the findings. To further explore the sources of heterogeneity, meta-regression analyses will also be conducted, variables such as sample size, publication year, and percentage of female participants will be taken into account.

Additionally, sensitivity analysis will also be used to test whether the pooled results are reliable. Sensitivity analyses will be conducted using the leave-one-out method to evaluate the influence of each study. Finally, the evidence quality will be assessed by the GRADE approach.

## Discussion

Many studies have investigated the correlation between sleep disorders exposure and POD outcomes [5,21]. Sleep disorders are prevalent and common during the perioperative period, and may increase the risk of POD. The objective of this study is to elucidate the association between sleep disorders and the risk of POD in adult patients. The findings of this review and meta-analysis will provide important data for future studies on intervention or treatment. Also, according to the results of the review and meta-analysis, patients with POD may take sleep disorders into account when making health decisions.

Key limitations of this review need to be considered. First, the different assessment methods of sleep disorders and the different definitions of POD can lead to a higher risk of publication bias. Therefore, subgroup analysis and meta-regression will be used to explore sources of heterogeneity. Second, it may be hard to establish causal relationships because most of the studies are expected to be observational. The GRADE tool will serve as a reference in the compilation of the summary table for the included studies. In general, our findings will help to improve the outcomes of patients with POD and provide better guidance for patients undergoing surgery.

## Supporting information

**S1 File.  PRISMA-P 2015 Checklist.**
(DOCX)

**S2 File.  Search strategies for included databases.**
(DOCX)

## Author contributions

**Conceptualization:** Qianli Huang, Changhui Shao, Wei Wei.

**Data curation:** Qianli Huang, Changhui Shao.

**Formal analysis:** Changhui Shao.

**Funding acquisition:** Wei Wei.

**Investigation:** Ling Peng.

**Methodology:** Qianli Huang, Ling Peng.

**Software:** Qianli Huang, Changhui Shao.

**Supervision:** Ling Peng.

**Writing – original draft:** Qianli Huang, Changhui Shao.

**Writing – review & editing:** Qianli Huang, Changhui Shao, Wei Wei.

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
