## [Decision Letter · Decision Letter 0]

9 Jul 2025

Dear Dr. Wei,

We look forward to receiving your revised manuscript.

Kind regards,

Jiawen Deng

Academic Editor

PLOS ONE

Journal Requirements:

[The research work was funded by a grant from the National Natural Science Foundation of China (grant no. 81971772).]. 

3. Thank you for stating the following in your manuscript:

[The research work was funded by a grant from the National Natural Science Foundation of China (grant no. 81971772). The funders had no role in study design, data collection and analysis, decision to publish, or preparation of the manuscript.]

[The research work was funded by a grant from the National Natural Science Foundation of China (grant no. 81971772).]

Reviewers' comments:

Reviewer's Responses to Questions

**Comments to the Author**

1. Does the manuscript provide a valid rationale for the proposed study, with clearly identified and justified research questions?

Reviewer #1: Yes

Reviewer #2: Yes

2. Is the protocol technically sound and planned in a manner that will lead to a meaningful outcome and allow testing the stated hypotheses?

Reviewer #1: Partly

Reviewer #2: No

3. Is the methodology feasible and described in sufficient detail to allow the work to be replicable?

Reviewer #1: Yes

Reviewer #2: Yes

4. Have the authors described where all data underlying the findings will be made available when the study is complete?

Reviewer #1: Yes

Reviewer #2: No

5. Is the manuscript presented in an intelligible fashion and written in standard English?

Reviewer #1: Yes

Reviewer #2: Yes

You may also provide optional suggestions and comments to authors that they might find helpful in planning their study.

Reviewer #1: Thank you for this protocol manuscript, which addresses an important topic. It is well written.

However, I do have some questions/feedback:

Pg 3: Financial disclosure statement missing

Pg 6 ln 120-121: ? does this include pre, peri and post op period? or just the perioperative period? Please clarify as earlier on you are presenting data from all periods?

Pg 6 ln 123: ?This is not an outcome. You need to define what outcomes you are specifically measuring - e.g. incidence of POD

Pg 6 ln 124-125: If you are looking for an association between the exposure and outcome, then why are you including cross sectional studies? Also, you discussed conflicts within literature between observational studies and RCT, yet you are excluding RCT - please justify your reasoning behind this.

Pg 6 ln 128-19: the use of insomnia symptoms or poor sleep quality are subjective measures, and thus if you are going to include them you need to clearly define what is meant by these. Also, how is poor sleep quality being identified? Using well defined criteria/diagnoses would provide a more objective and standard/consistent measure for sleep disorders.

Pg 6 ln 132: Also RCT according to your earlier criteria.

Pg 7 ln" 134: I am curious why Embase is being used, given its primary focus is on pharmaceuticals/drugs?

What about alternative databases, like CINAHL, especially as you indicate you are only including observational studies?

Pg 16 ln 194-199: what about the pre-existence of cognitive impairment/dementia, which is an important influencing factor, that can also impact on the diagnosis of POD?

Reviewer #2: Thank you for submitting this protocol on an important topic.

I have some concerns about the methodology you propose to study this topic. You indicate that your search will be limited to observational studies; however, you plan to search Cochrane Library, which indexes systematic reviews (CDSR) and randomized controlled trials (Central). This seems contradictory. Similarly, you justify your study by noting that since the publication of prior reviews there have been additional studies published which may contribute to further understanding; however, both of the additional studies cited appear to concern preoperative sleep disturbances, not postoperative.

Your search strategy also has issues that limit the methodological quality of the proposed review. You are missing potential synonyms such as emergence agitation and post surgical (with space). There are also unexplained differences in free-text search terms between different databases. Suggest also considering a search of PsycINFO, given the topic area.

**Do you want your identity to be public for this peer review?** For information about this choice, including consent withdrawal, please see our Privacy Policy

Reviewer #1: No

Reviewer #2: No

---

## [Author Response · Author response to Decision Letter 1]

18 Aug 2025

Dear Editor and Reviewers

Thank you very much for giving us the opportunity to revise our manuscript (PONE-D-25-29095). We sincerely appreciate the valuable comments and suggestions provided by the reviewers, which have significantly helped us to improve the quality of our manuscript. We have addressed each of the reviewers’ comments in detail below.

Reply: We have carefully checked throughout the manuscript.

[The research work was funded by a grant from the National Natural Science Foundation of China (grant no. 81971772).].

Reply: The funders had no role in study design, data collection and analysis, decision to publish, or preparation of the manuscript.

[The research work was funded by a grant from the National Natural Science Foundation of China (grant no. 81971772).]

Reply: we have removed any funding-related text from the manuscript and included our amended statements within cover letter.

Comments to the Author

Reviewer #1:

1. Pg 3: Financial disclosure statement missing

Reply: Thank you for your suggestion. We have added the financial disclosure statement.

2. Pg 6 ln 120-121: ? does this include pre, peri and post op period? or just the perioperative period? Please clarify as earlier on you are presenting data from all periods?

Reply: We have modified the description of the perioperative sleep disorders. The timing of sleep disorders includes either the pre- or post-operative period prior to delirium.

3. Pg 6 ln 123: ?This is not an outcome. You need to define what outcomes you are specifically measuring - e.g. incidence of POD

Reply: Thank you very much for your valuable comments, we have revised the outcomes.

4. Pg 6 ln 124-125: If you are looking for an association between the exposure and outcome, then why are you including cross sectional studies? Also, you discussed conflicts within literature between observational studies and RCT, yet you are excluding RCT - please justify your reasoning behind this.

Reply: Thank you very much for pointing out this issue. Given that cross-sectional studies cannot determine the association between the exposure and outcome, we have excluded the cross-sectional studies. In recent years, several observational studies have explored the association between sleep disorders and POD; however, few newly published RCTs have addressed this topic, so this review could not provide new insights into the association between sleep disorders and POD based on RCTs.

5. Pg 6 ln 128-19: the use of insomnia symptoms or poor sleep quality are subjective measures, and thus if you are going to include them you need to clearly define what is meant by these. Also, how is poor sleep quality being identified? Using well defined criteria/diagnoses would provide a more objective and standard/consistent measure for sleep disorders.

Reply: Thank you very much for your valuable comments. We have provided a more objective and standard/consistent measure for sleep disorders.

6. Pg 6 ln 132: Also RCT according to your earlier criteria.

Reply: Thank you for your suggestions. We have added this information in the exclusion criteria.

7. Pg 7 ln" 134: I am curious why Embase is being used, given its primary focus is on pharmaceuticals/drugs?

What about alternative databases, like CINAHL, especially as you indicate you are only including observational studies?

Reply: Thank you for your suggestions. We have added the CINAHL database for a comprehensive search strategy according to the suggested idea.

8. Pg 16 ln 194-199: what about the pre-existence of cognitive impairment/dementia, which is an important influencing factor, that can also impact on the diagnosis of POD?

Reply: Thank you very much for this good question. We have revised the participants of inclusion criteria.

Reviewer #2:

1. I have some concerns about the methodology you propose to study this topic. You indicate that your search will be limited to observational studies; however, you plan to search Cochrane Library, which indexes systematic reviews (CDSR) and randomized controlled trials (Central). This seems contradictory. Similarly, you justify your study by noting that since the publication of prior reviews there have been additional studies published which may contribute to further understanding; however, both of the additional studies cited appear to concern preoperative sleep disturbances, not postoperative.

Reply: Thank you very much for this good question. We have deleted the Cochrane Library. We have added this information in the manuscript as “However, since the publication of the previous meta-analysis, there have been additional studies published with large-scale investigations[15, 16]. Considering the importance of the topic, the aim of this study is to update the current evidence on the relationship between sleep disorders and the risk of POD.” The cited study, "Association of sleep quality on the night of the operative day with postoperative delirium in elderly patients: A prospective cohort study," explored the association between postoperative sleep disturbances and POD.

2. Your search strategy also has issues that limit the methodological quality of the proposed review. You are missing potential synonyms such as emergence agitation and post surgical (with space). There are also unexplained differences in free-text search terms between different databases. Suggest also considering a search of PsycINFO, given the topic area.

Reply: Thank you for pointing this out. We have added keywords such as "emergence agitation" and "post-surgical," as well as the PsycINFO database.

In summary, we have incorporated all the reviewers’ suggestions to the best of our ability, and we believe the revised manuscript has been significantly improved as a result. A detailed list of changes is highlighted in the revised manuscript.

---

## [Decision Letter · Decision Letter 1]

15 Oct 2025

Dear Dr. Wei,

Thank you for submitting your manuscript to PLOS ONE. After careful consideration, we feel that it has merit but does not fully meet PLOS ONE’s publication criteria as it currently stands. Therefore, we invite you to submit a revised version of the manuscript that addresses the points raised during the review process.

We look forward to receiving your revised manuscript.

Kind regards,

Giovanni Giordano

Academic Editor

PLOS ONE

Journal Requirements:

Reviewers' comments:

Reviewer's Responses to Questions

**Comments to the Author**

1. Does the manuscript provide a valid rationale for the proposed study, with clearly identified and justified research questions?

Reviewer #1: Yes

Reviewer #3: Yes

Reviewer #4: Yes

2. Is the protocol technically sound and planned in a manner that will lead to a meaningful outcome and allow testing the stated hypotheses?

Reviewer #1: Yes

Reviewer #3: Yes

Reviewer #4: Yes

3. Is the methodology feasible and described in sufficient detail to allow the work to be replicable?

Reviewer #1: Yes

Reviewer #3: Yes

Reviewer #4: Yes

4. Have the authors described where all data underlying the findings will be made available when the study is complete?

Reviewer #1: Yes

Reviewer #3: Yes

Reviewer #4: Yes

5. Is the manuscript presented in an intelligible fashion and written in standard English?

Reviewer #1: Yes

Reviewer #3: Yes

Reviewer #4: Yes

You may also provide optional suggestions and comments to authors that they might find helpful in planning their study.

Reviewer #1: Thank you for taking into consideration the reviewers comments, which you have incorporated into your revised manuscript. There are still a few questions I still have, for your review.

Reviewer #3: The revised manuscript, “The relationship between sleep disorders and postoperative delirium in adult patients: protocol for an updated systematic review and meta-analysis,” represents a substantial improvement over the previous version.

The authors have carefully addressed the reviewers’ main concerns, providing clearer methodological justifications, improving the definition of key variables, and aligning the study design with PRISMA-P standards.

The protocol is now methodologically sound, well structured, and scientifically relevant, especially given the ongoing clinical interest in postoperative delirium and perioperative sleep disorders.

Overall, the paper is suitable for publication after minor corrections.

The study aims and PECOS framework are now clearly described. It might still be helpful to slightly rephrase the objective sentence in the Objectives and outcomes section for smoother readability (e.g., “The objective of this review is to determine whether perioperative sleep disorders are associated with an increased risk of postoperative delirium in adult surgical patients.”).

The authors have adequately justified the exclusion of cross-sectional studies and randomized trials. However, the Data synthesis section could benefit from a brief statement explaining how discrepancies between subjective and objective measures of sleep quality will be handled during analysis, to further strengthen methodological clarity.

The Discussion provides a reasonable summary, though a few sentences could be edited for conciseness. In particular, lines describing the implications for “patients with POD making better health decisions” could be rephrased to focus on how clinicians might use these findings to improve perioperative management and patient outcomes.

The manuscript would benefit from a light linguistic edit to correct minor grammatical inconsistencies (e.g., missing articles, plural forms, and slight redundancies). This is not substantial but would enhance overall readability.

Ensure consistent use of tense (present vs. future) throughout the Methods section and uniform spacing between subsections.

The reference list is comprehensive and current.

Reviewer #4: Dear Editors and Authors,

I thank you for the opportunity to review the manuscript PONE-D-25-29095R1 titled “The relationship between sleep disorders and postoperative delirium in adult patients: protocol for an updated systematic review and meta-analysis”.

I have carefully reviewed the revised version of the manuscript, the supplemental files, and the detailed responses to the reviewers’ comments. Overall, the authors have made substantial improvements in addressing the concerns raised during the initial review.

Positive aspects:

• The clarification regarding the inclusion of pre- and post-operative periods in the assessment of sleep disorders adds important context to the study design.

• The refinement of outcome definitions, particularly specifying the incidence of POD, enhances the focus and clarity of the analysis.

• The exclusion of cross-sectional studies and the rationale for not including RCTs are now well justified by the authors.

• The inclusion of additional databases such as CINAHL and PsycINFO, strengthens the comprehensiveness of the systematic search strategy.

• Finally, the revision of inclusion criteria to consider cognitive impairment as a potential confounding factor appropriately addresses an important variable influencing POD diagnosis.

Suggestions for further improvement:

• The addition of the keyword "emergence agitation" and “post surgical” in the search strings of the various databases is a positive step. In the Search Strategy paragraph, I recommend also including the term "post surgical," as suggested by Reviewer 2.

• Furthermore, I believe it would be useful to specify the use of Boolean operators (AND, OR) in the Search Strategy paragraph as well, in order to clarify how the different groups of keywords are connected in the search.

Minor grammatical and syntactic improvements:

• Line 33: “Pubmed” → “PubMed”

• Line 39: “used” → “conducted”

• Line 68: “a” after “is,” not before (typographical error)

• Line 86: “had investigated association” → “had investigated the association” (add “the”)

• Line 106: “is summary and analysis” → “is a summary and analysis” (missing article)

• Line 112: “The sole objective of this is to determine” → “The sole objective of this study is to determine” (add “study” for clarity)

• Line 114: “Compares” → In the PECOS model, C stands for “Comparator,” not “Compares”

• Line 132: “association of sleep disorders and the risk of POD” → “association between sleep disorders and the risk of POD”

• Lines 132-133: “Included participants diagnosed with sleep disorders and no sleep disorders.” → “Studies including participants with and without diagnosed sleep disorders.”

• Line 133: “subjectively or objectively diagnostic criteria” → “subjective or objective diagnostic criteria”

• Line 143: Remove comma after “2025”

• Line 155: “in The Endnote X9 software reference manager” → “in EndNote X9 reference management software”

• Line 169: “contacted for obtain the relevant information” → “contacted to obtain the relevant information”

• Line 173: Avoid repeating “scale” twice in “The Newcastle Ottawa Scale (NOS) assessment scale will be used…”

• Line 174: “researches” → “studies” (in scientific English “research” is uncountable, use “studies or research studies” in plural)

• Line 192: “otherwise narrative synthesis will be taken” → “otherwise a narrative synthesis will be carried out”

• Lines 197: “was evaluated” → Should be future tense: “will be evaluated”

• Lines 217-219: “Sleep disorders are prevalent and common during the surgery” → I think that “during the perioperative period” is more appropriate

• Lines 219-220: “association of sleep disorders and the risk of POD” → “association between sleep disorders and the risk of POD”

• Lines 222-223: “patients with POD may take into consideration the sleep disorders to make better health decisions” → “patients with POD may take sleep disorders into account when making health decisions”

• Line 225: “higher risk of publication” → Add “bias”: “higher risk of publication bias”

• Lines 227: “should be used to explore the source of heterogeneity” → “will be used to explore sources of heterogeneity”

Overall recommendation:

The manuscript shows substantial enhancement and better alignment with PLOS ONE’s publication criteria. Pending minor revisions as outlined, I believe it merits acceptance for publication. The authors are encouraged to refine the indicated sections to further improve transparency and reproducibility.

I thank the authors for their commitment to improving the manuscript and remain available for any further review if needed.

Best regards,

Gaetano Gazzè

**Do you want your identity to be public for this peer review?** For information about this choice, including consent withdrawal, please see our Privacy Policy

Reviewer #1: No

Reviewer #3: No

Reviewer #4: **Yes: ** Gaetano Gazzè

---

## [Author Response · Author response to Decision Letter 2]

5 Nov 2025

Dear Editor and Reviewers:

We would like to thank the Editors and Reviewers for their thoughtful and constructive comments. We have addressed each point below. All changes are indicated in the tracked version of the revised manuscript and are described below in a point-by-point response.

Suggestions for further improvement:

1. The addition of the keyword "emergence agitation" and “post surgical” in the search strings of the various databases is a positive step. In the Search Strategy paragraph, I recommend also including the term "post surgical," as suggested by Reviewer 2.

Response: Thank you for your suggestion. We have added the term “post surgical” as suggested.

2. Furthermore, I believe it would be useful to specify the use of Boolean operators (AND, OR) in the Search Strategy paragraph as well, in order to clarify how the different groups of keywords are connected in the search.

Response: We have added Boolean operators (“AND” and “OR”) in the Search Strategy paragraph to combine search terms.

3. Minor grammatical and syntactic improvements:

•Line 33: “Pubmed” → “PubMed”

Response: We appreciate your careful reading of our manuscript. We have revised as suggested.

• Line 39: “used” → “conducted”

Response: We have revised as suggested.

• Line 68: “a” after “is,” not before (typographical error)

Response: We have revised as suggested.

• Line 86: “had investigated association” → “had investigated the association” (add “the”)

Response: We have added “the” as suggested.

• Line 106: “is summary and analysis” → “is a summary and analysis” (missing article)

Response: We have added “a” as suggested.

• Line 112: “The sole objective of this is to determine” → “The sole objective of this study is to determine” (add “study” for clarity)

Response: We have added “study” as suggested.

• Line 114: “Compares” → In the PECOS model, C stands for “Comparator,” not “Compares”

Response: We have revised as suggested.

• Line 132: “association of sleep disorders and the risk of POD” → “association between sleep disorders and the risk of POD”

Response: We have revised as suggested.

• Lines 132-133: “Included participants diagnosed with sleep disorders and no sleep disorders.” → “Studies including participants with and without diagnosed sleep disorders.”

Response: We have revised as suggested.

• Line 133: “subjectively or objectively diagnostic criteria” → “subjective or objective diagnostic criteria”

Response: We have revised as suggested.

• Line 143: Remove comma after “2025”

Response: We have revised as suggested.

• Line 155: “in The Endnote X9 software reference manager” → “in EndNote X9 reference management software”

Response: We have revised as suggested.

• Line 169: “contacted for obtain the relevant information” → “contacted to obtain the relevant information”

Response: We have revised as suggested.

• Line 173: Avoid repeating “scale” twice in “The Newcastle Ottawa Scale (NOS) assessment scale will be used…”

Response: We have revised as suggested.

• Line 174: “researches” → “studies” (in scientific English “research” is uncountable, use “studies or research studies” in plural)

Response: We have revised as suggested.

• Line 192: “otherwise narrative synthesis will be taken” → “otherwise a narrative synthesis will be carried out”

Response: We have revised as suggested.

• Lines 197: “was evaluated” → Should be future tense: “will be evaluated”

Response: We have revised as suggested.

• Lines 217-219: “Sleep disorders are prevalent and common during the surgery” → I think that “during the perioperative period” is more appropriate

Response: We have revised as suggested.

• Lines 219-220: “association of sleep disorders and the risk of POD” → “association between sleep disorders and the risk of POD”

Response: We have revised as suggested.

• Lines 222-223: “patients with POD may take into consideration the sleep disorders to make better health decisions” → “patients with POD may take sleep disorders into account when making health decisions”

Response: We have revised as suggested.

• Line 225: “higher risk of publication” → Add “bias”: “higher risk of publication bias”

Response: We have added “bias” as suggested.

• Lines 227: “should be used to explore the source of heterogeneity” → “will be used to explore sources of heterogeneity”

Response: We have revised as suggested.

4. Overall recommendation:

The manuscript shows substantial enhancement and better alignment with PLOS ONE’s publication criteria. Pending minor revisions as outlined, I believe it merits acceptance for publication. The authors are encouraged to refine the indicated sections to further improve transparency and reproducibility.

Response: We are truly grateful for your thoughtful evaluation and constructive feedback, which have greatly contributed to the improvement of our manuscript.

---

## [Decision Letter · Decision Letter 2]

11 Nov 2025

The relationship between sleep disorders and postoperative delirium in adult patients: protocol for an updated systematic review and meta-analysis

PONE-D-25-29095R2

Dear Dr. Wei,

We’re pleased to inform you that your manuscript has been judged scientifically suitable for publication and will be formally accepted for publication once it meets all outstanding technical requirements.

Kind regards,

Giovanni Giordano

Academic Editor

PLOS ONE

Additional Editor Comments (optional):

Reviewers' comments:

Reviewer's Responses to Questions

**Comments to the Author**

1. Does the manuscript provide a valid rationale for the proposed study, with clearly identified and justified research questions?

Reviewer #4: Yes

2. Is the protocol technically sound and planned in a manner that will lead to a meaningful outcome and allow testing the stated hypotheses?

Reviewer #4: Yes

3. Is the methodology feasible and described in sufficient detail to allow the work to be replicable?

Reviewer #4: Yes

4. Have the authors described where all data underlying the findings will be made available when the study is complete?

Reviewer #4: No

5. Is the manuscript presented in an intelligible fashion and written in standard English?

Reviewer #4: Yes

You may also provide optional suggestions and comments to authors that they might find helpful in planning their study.

Reviewer #4: Good morning,

I believe the protocol has now reached a solid and satisfactory standard. I wish you the best of luck with the remaining work to complete the study.

Kind regards,

Gaetano Gazzè

**Do you want your identity to be public for this peer review?** For information about this choice, including consent withdrawal, please see our Privacy Policy

Reviewer #4: **Yes: ** Gaetano Gazzè

---

## [Editor Report · Acceptance letter]

PONE-D-25-29095R2

PLOS ONE

Dear Dr. Wei,

I'm pleased to inform you that your manuscript has been deemed suitable for publication in PLOS ONE. Congratulations! Your manuscript is now being handed over to our production team.

Kind regards,

on behalf of

Dr. Giovanni Giordano

Academic Editor

PLOS ONE